# *Gentianella lutescens* subsp. *carpatica* J. Holub.: Shoot Propagation In Vitro and Effect of Sucrose and Elicitors on Xanthones Production

**DOI:** 10.3390/plants10081651

**Published:** 2021-08-11

**Authors:** Dijana Krstić-Milošević, Nevena Banjac, Teodora Janković, Dragan Vinterhalter, Branka Vinterhalter

**Affiliations:** 1Department of Plant Physiology, Institute for Biological Research “Siniša Stanković”—National Institute of the Republic of Serbia, University of Belgrade, Bulevar despota Stefana 142, 11000 Belgrade, Serbia; mitic.nevena@ibiss.bg.ac.rs (N.B.); dvinterhalter@yahoo.com (D.V.); horvat@ibiss.bg.ac.rs (B.V.); 2Insitute for Medicinal Plants Research “Dr Josif Pančić”, Tadeuša Košćuška 1, 11000 Belgrade, Serbia; tjankovic@mocbilja.rs

**Keywords:** shoot culture, secondary metabolites, HPLC, bellidifolin, osmotic stress

## Abstract

In vitro shoot culture of the endangered medicinal plant *Gentianella lutescens* was established from epicotyl explants cultured on MS basal medium with 0.2 mg L^−1^ 6-benzylaminopurine (BA) and evaluated for xanthones content for the first time. Five shoot lines were obtained and no significant variations in multiplication rate, shoot elongation, and xanthones profile were found among them. The highest rooting rate (33.3%) was achieved by shoots treated for 2 days with 5 mg L^−1^ indole-3-butyric acid (IBA) followed by cultivation in liquid PGR-free ½ MS medium for 60 days. HPLC analysis revealed the lower content of xanthones—mangiferin, bellidifolin, demethylbellidifolin, demethylbellidifolin-8-*O*-glucoside and bellidifolin-8-*O*-glucoside—in in vitro cultured shoots compared to wild growing plants. The increasing concentration of sucrose, sorbitol and abiotic elicitors salicylic acid (SA), jasmonic acid (JA) and methyl jasmonate (MeJA) altered shoot growth and xanthone production. Sucrose and sorbitol applied at the highest concentration of 233.6 mM increased dry matter percentage, while SA at 100 μM promoted shoot growth 2-fold. The increased sucrose concentration enhanced accumulation of xanthones in shoot cultures 2–3-fold compared to the control shoots. Elicitors at 100–300 μM increased the accumulation of mangiferin, demethylbellidifolin-8-*O*-glucoside, and bellidifolin-8-*O*-glucoside almost equally, while MeJA at the highest concentration of 500 μM enhanced amount of aglycones demethylbellidifolin and bellidifolin 7-fold compared to the control. The obtained results facilitate conservation of *G. lutescens* and pave the way for further research on large-scale shoot propagation and production of pharmacologically active xanthones.

## 1. Introduction

The genus *Gentianella* Moench (Gentianeceae) encompasses about 250 species growing mainly in temperate or mountain habitats in Europe, South America, New Zealand, and Australia [1]. In Europe, the genus *Gentianella* consists of 22 species with numerous subspecies and taxa that are distributed in the Alps, Carpathians, in the Tatra mountains, as well as in the mountains of the Balkan Peninsula [2]. In Serbia, the genus *Gentianella* is represented by six species: *G. austriaca*, *G. bulgarica*, *G. axilaris*, *G. ciliata*, *G. praecox*, and *G. crispata* [3].

Apart their ornamental value, *Gentianella* species have been well-known as traditional medicines since ancient times. In South America they have been used as a traditional remedy for the treatment of digestive and liver problems [4,5]. *Gentianella amarella* is known in the traditional medicine of Mongolia to cure headache, hepatitis, fever, and gallbladder disorders [6]. Like other species of the Gentianaceae family, *Gentianella* plants are characterized by the universal occurrence of three main groups of secondary metabolites, iridoids, flavonoids, and xanthones [7].

Naturally occurring xanthones have been attracting attention for a long time due to their specific bioactivities and occupy an important position in the pharmacology and chemistry of natural products. Xanthone compounds typical of *Gentianella* species belong to the bellidifolin type of xanthones, which mostly occur in the form of *O*-glycosides. They are responsible for wide range of therapeutic properties attributed to *Gentianella* plants. The xanthones bellidifolin and demethylbellidifolin, the principal constituents of many gentianellas, have been reported to show cardioprotective effects, antioxidant, antimicrobial, and antidiabetic activity, as well as displaying significant potential to inhibit acetylcholinesterase and monoamino oxidase activity [8]. Bellidifolin exhibited a variety of pharmacological activities, including prominent hypoglycemic [9] and neuroprotective activities [10]. These findings support the potential use of xanthone compounds as new drugs in treating aging-related neurodegenerative disorders [11] and also as useful candidates for therapy of type 2 diabetes [9].

Although finding *Gentianella* species as a reach source of important phytochemicals has heightened research interest in these species, many of them have not yet been properly investigated due to the low availability of plant material. A large number of *Gentianella* species are rare and endemic or grow in inaccessible localities, while the others became endangered from excessive harvesting or adverse environmental conditions. Such an unfavorable situation is the case with the endemic species *G. lutescens* subsp. *carpatica*, which we discovered in Serbia for the first time during field research on Povlen mountain (Figure 1). A survey of literature indicated that this species has not been phytochemically investigated so far and very scarce data can be found about it.

It is biennial 3–40-cm-tall plant, and is simple or branched in the upper part, forming a racemose inflorescence panicle-like, umbrella-shape with reddish to violet flowers. *Gentianella lutescens* subsp. *carpatica* was located in Eastern and Central Europe and northern part of the Balkan Peninsula, mainly in the mountains, with ecotypic variants in Austria, Bulgaria, Czech, Germany, Yugoslavia, and Poland [12]. This species was reported as relatively common in the Czech Republic before 1950; however, nowadays it is considered a critically threatened plant surviving on only a few sites [13]. The fact that the extant populations are small, often less than ten individuals, signify that *G. lutescens* is a critically endangered plant species that is included in both the European and world red list, according to International Union for Conservation of Nature’s (IUCN) Red List [14].

Advances in plant biotechnology and tissue culture have proven to be a valuable tool for large-scale propagation, storage, reintroduction, as well as the production of secondary metabolites of endangered medicinal plant species [15,16,17].

Many papers have been published over the last two decades related to in vitro culture studies of more eminent members of Gentianaceae, such as *Gentiana* spp. and *Swertia* spp. However, these studies were performed in only a few species of the genus *Gentianella*, and included *G. austriaca* [18], *G. bulgarica* [19,20], *Gentianella albifpra* [21], and *G. bicolor* [22]. Considering the rarity and medicinal importance of *G. lutescens,* tissue culture would be a suitable tool, not only for conservation purposes, but also for overcoming the deficit of plant material in order to perform phytochemical investigations and in the development of technology for the extraction of important metabolites [23]. Although a lower content of secondary metabolites of interest in the tissue of cultured plants compared to natural plants may limit the applicability of tissue culture [17], it also offers the possibility to stimulate their production using various strategies.

Elicitation with chemical compounds is an effective strategy for improving the production of secondary metabolites in plant cells and tissue cultures. Elicitor molecules trigger signaling events leading to the activation of defense-related genes involved in the biosynthesis of secondary metabolites. Increased transcription of genes encoding biosynthetic enzymes of central phenylpropanoid pathway (phenyilalanine-ammonia lyase (PAL) and 4-coumarate-CoA-ligase (4 CL)) and specific enzyme benzophenone synthase (BPS) of the xanthone synthesis branch route has been demonstrated upon elicitation of *Hypericum perforatum* cell cultures [24]. Salicylic acid (SA), jasmonic acid (JA), and methyl jasmonate (MeJA) are plant defense signaling molecules, and, when exogenously added, they induce systemic acquired resistance against stress, thus acting as elicitors. Several studies reported higher production of xanthones and flavonoids in cell suspensions of *H. perforatum* treated with SA, JA, or MeJA [25,26,27]. Exposure to high levels of sucrose or sorbitol have also induced an increased accumulation of some secondary metabolites caused by osmotic stress [28,29].

In view of medicinal and conservation importance of *G. lutescens*, the aim of the current work was to evaluate in vitro grown shoot culture of *G. lutescens* as an alternative, sustainable, and stable source of xanthones. In order to increase biomass and xanthones content in *G. lutescens* shoots, the effect of sucrose, sorbitol, and elicitors JA, MeJA, and SA, on shoot growth and xanthones production was investigated.

## 2. Results and Discussion

### 2.1. HPLC-DAD Analysis of Secondary Metabolites of Wild Grown G. lutescens Plants

The chemical profile of methanol extract of aerial parts of wild-growing *G. lutescens* plants analyzed using the HPLC-DAD technique is presented in Figure 2. Similarly, as in other *Gentianella* species, three groups of secondary metabolites, secoiridoids, flavone-*C*-glucosides, and xanthones, were detected. Chromatographic analysis identified the presence of swertiamarin and gentiopicrin (peaks 1 and 2, respectively) as the most common secoiridoid compounds, which appeared to be present in all species of Gentianaceae [30].

Peaks 5 and 7 were identified as isoorientin and swertisin, the most represented *C*-glucoflavones in the *Gentianella* species. The seven remaining peaks presented in the chromatogram were detected as xanthone compounds. The precise identification of each xanthone was confirmed using the HPLC co-injection method using reference xanthone standards, previously isolated in our laboratory [31]. The two dominant peaks (Figure 2) belong to demethylbellidifolin-8-*O*-glucoside (6) and bellidifolin-8-*O*-glucoside (8), xanthones with 1,3,5,8-oxidation pattern characteristic for *Gentianella* species. HPLC also revealed the presence of a tetrahydroxanthone glucoside named campestroside (4), a partially saturated analog of demethylbellidifolin-8-*O*-glucoside. A xanthone with such a structure need special attention since it is rare in nature and its occurrence is of particular chemotaxonomic and biogenetic significance [31]. Compound 9 was identified as xanthone-*O*-glucoside veratriloside, and this compound was reported to be the first 1,3,4,7-oxygenated xanthone isolated from the genus *Gentianella* [20,31]. Peak 3 belongs to *C*-glucoxanthone mangiferin, one of the well-known naturally occurring xanthones, widespread among angiosperms. The occurrence of mangiferin together with flavone-*C*-glucosides isoorientin and swertisin is most common and is typical for *Gentianella* species. The peaks detected at the end of the chromatogram were identified as xanthone aglycons demethylbellidifolin (10) and bellidifolin (11). Figure 3 shows the chemical structures of the secoiridoid and xanthone compounds identified in *G. lutescens*.

Considering that xanthones are becoming increasingly important compounds that possess a broad spectrum of biological and pharmacological activities, in this study we analyzed secondary metabolites from *G. lutescens* cultured in vitro. Specifically, we focused on the chemical analysis of five dominant xanthones: mangiferin, demethylbellidifolin-8-*O*-glucoside (DMB-8-*O*-glc), bellidifolin-8-*O*-glucoside (bell-8-*O*-glc), and the aglycons demethylbellidifolin (DMB) and bellidifolin.

### 2.2. In Vitro Shoot Propagation of G. lutescens

Immature seeds of *G. lutescens* (Figure 4A) were germinated for 10 days at a rate of 5%, and non-contaminated seeds were found. The maximum response of the epicotyl explants to produce new shoots was observed after 35 days of culture on shoot induction medium (Figure 4B). The genotype of the individual seedlings did not have a significant effect on the shoot proliferation response, as a multiplication index of about 3 was recorded in all five lines. However, the genotype significantly affected the elongation of the main shoot, ranging from 14.49 mm in line 1 to 23.05 mm in line 3 (Table 1).

Elongated shoots (≥15 mm) sporadically formed flower buds. The shoots with intense blue–violet flowers developed normally on the cytokinin-containing medium. With the aim to enhance shoot proliferation, the individual shoots of line 5 were transferred onto BM with increasing concentrations of BA (0–2.0 mg L^−1^). Line 5 was chosen for the shoot proliferation experiment because it showed growth and multiplication stability over time and produced a large number of healthy-looking shoots that were needed to start the experiment. According to ANOVA, shoot multiplication and shoot length were significantly affected by the concentration of BA (Table 2). A higher BA concentration (above 0.5 mg L^−1^), not only contributed to the enhancement of shoot multiplication rate, but also increased the percent of low-quality vitrified shoots (Table 2). Moreover, the length of the main shoot was significantly decreased with increasing BA concentration. The highest multiplication rate, close to 4, was achieved on basal medium supplemented with 0.2 mg L^−1^ BA along with minimum reduction of the main shoot length compared to cytokinin-free medium. In light of this finding, and considering the reasonable percent of vitrified shoots (7.75%), the above medium formulation was selected as optimal for shoot culture multiplication and maintenance. Unexpectedly, a relatively satisfactory multiplication rate (2.89) was achieved on cytokinin-free medium relative to BA-containing media. This can be explained by prolonged effect of BA from shoot induction medium.

In cultures maintained on basal medium supplemented with 0.2 mg L^−^^1^ BA more than one year, the number of shoots with spontaneously developed flowering buds gradually decreased over time and became rare in those maintained more than 5 years. Cytokinin was a common requirement for in vitro flowering, which also occurred in *G. austriaca* [18] and *G. bulgarica* shoot cultures [19]. However, while precocious in vitro flowering in these two species threatened shoot multiplication at a higher rate due to decay of the main shoot after flowering, it appeared sporadically in *G. lutescens* and has not been observed to significantly affect multiplication. Nevertheless, in vitro flowering could serve as an important tool for many purposes. Manipulation of different variables in in vitro conditions offers a unique system for studying molecular basis and hormonal regulation of flowering. On the other hand, in vitro flowering could be a valuable tool for the faster release of new varieties [32], enabling also the recombination of genetic material via in vitro fertilization in otherwise non-hybridizable lines. Since the multifunctional usability of in vitro flowering, especially in rare and endangered plant species, such as *G. lutescens*, comprehensive study of this phenomenon could be the main goal of further research studies in *G. lutescens*.

To the best of our knowledge, this is the first report on the establishment of shoot cultures of endemic *G. lutescens*. In general, *Gentianella* species were, so far, barely investigated compared to their closest relatives belonging to the genus *Gentiana*. Huo and Zheng [21] were the first to reported shoot regeneration from calli of *G. albifpra* cultured on medium with 3.0 mg L^−1^ 2,4-D + 1.0 mg L^−1^ KIN. Later on, in vitro propagation was achieved from epicotyls of sterile germinated mature seeds of *G. austriaca* [18] and *G. bulgarica* [19], which are endemic in central part of the Balkan Peninsula. Incorporation of cytokinin promoted shoot explant proliferation in *Gentianella* species and BA was found to be superior for new shoot formation. BA was also effective for inducing shoot proliferation in the previously mentioned *Gentianella* species, *G. austriaca* and *G. bulgarica*, where BA at 0.2–0.5 mg L^−1^ (depending on species) with the addition 0.1 mg L^−1^ NAA was applied [18,19]. More recently, Solorzano et al. [22] reported shoot regeneration from leaf-explant-derived calli of *G. bicolor* on medium containing a combination of KIN and 2,4-D. However, shoot tips, epicotyl, and nodal segments are generally preferred explants for multiplication of most plant species due to the presence of pre-existing meristems. They can be easily developed into shoots that ensure clonal fidelity [33]. Genetic fidelity and true-to-type regenerated plants are very important for both the germplasm maintenance for plant conservation purposes and mass shoot propagation that ensures a continuous supply of uniform genetic plant material for large-scale secondary metabolite production [34,35].

Rooting of *G. lutescens* shoots was performed on a medium with a reduced content of MS mineral salts (by half (½ MS)), supplemented with increasing concentrations of IBA at 0.2–5.0 mg L^−1^ (Table 3). However, if only solid medium was used, auxin-induced root primordia did not elongate and redundant calli were observed, especially at IBA 0.5–2.0 mg L^−1^ (data not shown). Transfer of shoots with induced roots to solid auksin-free medium, with or without active charcoal (1%) or GA_3_ (0.1 mg L^−1^), also did not stimulate elongation of root primordia (data not shown). Elongation of IBA-induced roots was achieved by transferring shoots into liquid PGR-free ½ MS medium. Decades of work with gentians and other plant species suggest that increased concentrations of auxin should reduce the length of induction treatment. Otherwise calli will form instead of roots. Hence, in *G. lutescens*, we used different root induction treatments corresponding to auxin concentration (Table 3). Elongation of the roots in the liquid medium was very slow and required about 60 days. The best rooting was achieved by induction with IBA 5.0 mg L^−1^ for 2 days, followed by root elongation in liquid PGR-free ½ MS medium for 61 days. This protocol provided the highest percentage of rooted shoots (33%), number of shoots with induced root primordia (23.3), number of roots per rooted shoots (5.4), and root elongation (11.2 mm) (Table 3; Figure 4C). These results indicated *G. lutescens* as a highly recalcitrant species in terms of rooting potential that should be further improved to obtain enough healthy and functional plants for successful acclimatization and ex situ and in situ conservation. Other *Gentianella* species also displayed restricted rooting potential. The highest rooting percentage was obtained for *G. austriaca*, where 47.3% of shoots formed roots on solid MS medium with 4.92 μM (1.0 mg L^−1^) IBA [18]. However, only 1–2% shoots of *G. bulgarica* spontaneously rooted on plant growth regulator free medium [19].

### 2.3. HPLC-DAD Analysis of Secondary Metabolites of G. lutescens Cultured In Vitro

HPLC-DAD analysis of methanol extracts obtained from five shoot lines of *G. lutescens* showed no significant differences between them in either qualitative or in quantitative xanthone composition. It can be noticed that all 5 lines produced a lower content of xanthones compared to the plant material collected in nature (Figure 5). This was not surprising result since the relatively low number of in vitro plants surpassed wild plants in terms of the amounts of secondary metabolites [36].

According to ANOVA, cytokinin BA, necessary for shoot multiplication of *G. lutescens*, did not have a significant effect on the production of xanthones (Appendix A). However, a different effect of BA was reported in *G. bulgarica* shoots where xanthone production was strongly affected by BA. Namely, it has been shown that the content of xanthones linearly increased with increasing BA concentration (0.2–1.0 mg L^−1^) in medium [19]. The stimulatory effect of BA on secondary metabolite production has been shown in shoot cultures of *G. austriaca* (2.2–4.4 µM) [18] and *Gentiana asclepiadea* (4.4 µM) [37] as well.

### 2.4. The Effect of Sucrose and Sorbitol on Shoot Growth and Xanthone Production

According to ANOVA, increasing concentration of sucrose in the growth medium significantly affected shoot growth, dry matter, and flowering (Appendix A). Thus, additional sucrose, from 58.4 mM to 233.6 mM, gradually decreased shoot fresh weight and growth index (Figure 6A,C). At the same time, the dry weight of the shoots was not significantly affected, while dry matter percentage even significantly increased with rising sucrose concentrations (Figure 6B,D).

In plant tissue culture, sucrose serves as a carbon and energy source necessary for cell division and differentiation [38] and for the regulation of osmotic potential [39]. In the present study, the highest shoot fresh weight (1600 mg) and the growth index (3.2) were recorded in the control *G. lutescens* shoots grown at the lowest (58.4 mM) sucrose. According to Grattapaglia and Machado [40], sugar concentrations lower or higher than 58.42 mM can cause chlorosis or explant deterioration, respectively, in in vitro cultures. The increased concentration of sucrose induces osmotic stress due to enhanced osmotic potential in the growth medium, which inhibits shoots water and nutrients uptake from the medium [41]. This in turn significantly reduces shoots fresh weight and the growth index as compared to the dry weight. Osmotic potential may interfere with nutrient abortion by cells, which is essential to growth and cell division in the aerial parts [42]. This can partly explain the observations of reduced *G. lutescens* shoot fresh weights with sucrose concentrations increasing above 58.4 mM (Figure 6D). Increment of flowering was found (Figure 7) from 0 to 1.4 flowers per Erlenmeyer flask with increasing sucrose concentration from 58.4 in control to 175.2 mM, respectively, and then gradually decreased at 175.2 mM and 233.6 mM sucrose (1.2 and 0.4 flowers per Erlenmeyer flask, respectively) (data not shown) and could be also stress related phenomenon [43].

Additionally, the higher sucrose concentrations at 175.2 and 233.6 mM caused a color change of *G. lutescens* shoots from green to yellow–brown along with stunted growth (Figure 7). High sugar concentrations (45 and 60 g L^−1^) were found to inhibit the growth of aerial plant parts [44], with a reduction of photosynthetic pigments content [45] compared to plants grown on medium without sucrose, as was verified on *B. zebrina* shoots [46]. The reduction in chlorophyll content in in vitro plants may reduce photosynthetic ability by decreasing light absorption [47].

On the other hand, although the increased sucrose concentration in the medium above 58.4 mM was not optimal for *G. lutescens* shoot fresh weight and growth index, continuing increasing of dry matter percentage *G. lutescens* shoots over all higher ranges of sucrose tested was indicative (Figure 6). A positive correlation between sucrose in the medium and dry matter content in in vitro plants has been previously reported. Increased availability of sugars in heterotrophic systems has been shown to increase cellulose synthesis, which was correlated to increase in dry weight [48]. In addition, sucrose cleavage in the medium results in glucose and fructose production. It may accelerate cell division and consequently increase the explant weight and volume [49].

Numerous studies indicated sorbitol as the best carbon source for plant multiplication in many species [50,51], as well as to distinguish nutritive effect of enhanced sucrose in the growth medium from its osmotic effect. Here, we analyzed the relative impact of increased sorbitol concentrations on *G. lutescens* shoot cultures. According to ANOVA, increasing concentration of sorbitol significantly altered shoot fresh weight, growth index, dry matter percentage, but not dry weight (Appendix A).

Results of the present study revealed analogical effect of increasing concentrations of sorbitol (58.4–233.6 mM) with that of sucrose on *G. lutescens* shoot fresh weight, dry weight, and growth index (Figure 6). Moreover, the effect of enhanced sorbitol on increased dry matter percentage (Figure 6H) at an almost identical range compared to that of sucrose (Figure 6D) was observed.

Two-way ANOVA indicated that the type of carbohydrate added did not affect the growth of the shoots, but in vitro flowering and dry matter percentage were significantly affected. However, the concentrations of carbohydrates significantly affected all traits in the same trend (Appendix A).

These findings strongly confirmed osmotic effect of sucrose on *G. lutescens* and appointed 58.4 mM sucrose as optimal for shoots growth in vitro.

Regarding the effect of sucrose and sorbitol on the xanthone contents in *G. lutescens* shoots, the obtained results showed that increased sucrose concentration stimulated the production of xanthones, whereas the effect of sorbitol was weak or even absent (Figure 8, Appendix A). Namely, the level of either xanthone analyzed significantly increased with increasing sucrose in the medium and the highest content of xanthones was recorded at the higher sucrose concentration applied. Thus, the content of mangiferin at 233.6 mM of sucrose was more than 3-fold higher compared to shoots cultured on the control medium containing 58.4 mM of sucrose. The highest sucrose concentration enhanced the production of xanthone glucosides DMB-8-*O*-glc and bell-8-*O*-glc for 2.8- and 1.8-times, respectively. The amount of aglycon bellidifolin increased 2.8-times, while the accumulation of DMB increased more than 4 times compared to the control.

The results presented in Figure 8A–E show that xanthone content linearly increased with increasing sucrose concentrations, up to 116.8 mM. With additional increase in sucrose at 175.2 and 233.6 mM, the content of each xanthone reached its highest level. On the other hand, the content of xanthones in shoots cultured with sorbitol was mainly at the level recorded in the control shoots. Compared to the control, only the production of DMB-8-*O*-glc, DMB, and bellidifolin was slightly higher in shoots cultured with the lowest sorbitol concentration (Figure 8G–J).

These results indicate that enhanced accumulation of xanthones in shoots caused by increased concentrations of sucrose might be a consequence of the nutritional effect of sucrose. Numerous studies have shown that higher content of sucrose in the medium stimulated the production of useful secondary metabolites in plant cell cultures. Sucrose is considered to be one of the key sugars in plant life. In addition to its primary building role, sucrose is also an energy supplier for the production of plant biomass. It is also involved in growth, development, storage, signaling, plant stress responses and various metabolic processes [52]. The abundance of sucrose in the culturing medium quite certainly affects and alters metabolic activities in the shoots of *G. lutescens*. The accretion of available source of carbon and energy directs metabolic pathways to formation more complex compounds with a large carbon skeleton, such as xanthones. Further, xanthones may undergo hydroxylation reactions to give xanthone glycosides. As explained in our previous study on the root cultures of *Gentiana dinarica*, higher metabolic activities, due to the increase of available carbohydrates sources as biochemical substrates, may direct xanthone biosynthesis to the enhanced production of xanthone glycosides [28].

The similar positive effect of increased sucrose in the medium on the xanthone content has been reported in *Centaurium erythraea* cultured in vitro [53]. The higher sucrose concentration was more favorable for accumulation of phenols, flavonoids, chlorogenic acid, and total hypericin in adventitious root cultures of *Hypericum perforatum* [54]. The roots of *Gentiana dinarica* produced higher content of xanthones when cultured on medium with more sucrose [28].

### 2.5. The Effect of Elicitors JA, MeJA and SA on Shoot Growth and Xanthone Production

The elicitation process is indicated as very complex and is subject to many factors, such as elicitor concentration, shoot growth stage, as well as the time of plant tissue exposure to the elicitor [55,56].

In an attempt to enhance xanthones production, JA, MeJA and SA at increasing concentrations (100–500 µM) were applied in shoot culture of *G. lutescens*. Plant tissue responds relatively rapidly (from several hours to several days) to elicitation to induce the synthesis of secondary metabolites [57,58] while prolonged exposure to elicitor may reduce their accumulation over time. For example, the highest production of bacoside in shoots of *Bacopa monnieri* was obtained in the first 7 days of elicitation with MeJA [59] and JA [60]. Guided by the positive results of the elicitation protocol for *Withania somnifera* with SA and MeJA [61], shoot growth and xanthones amounts were analysed in *G. lutescens* shoots cultivated for 7 days on an elicitor-containing medium and another 7 days on an elicitor-free medium.

Figure 9A shows that growth index of the shoots significantly increased after SA low concentration (100 μM) treatment and significantly decreased after treatment with MeJA at the highest one (500 μM), whereas shoot growth was not affected by JA at all concentrations (Figure 9A, Appendix A). The positive effect of low concentration of SA on *G. lutescens* is not surprising. In general, low concentrations of applied SA promote plant growth under unfavorable conditions. In contrast, high SA concentrations inhibit growth, while the threshold between low and high concentrations depend on plant species. It was shown that SA exhibited growth-promoting (50 μM) and growth-inhibiting (250 μM) effect on *Matricaria chamomilla* seedlings [62].

SA, JA, and MeJA, were reported to elicit a wide spectrum of phytochemicals in different plant species by inducing the expression of genes for various biosynthetic pathways. Thus, MeJA was reported to stimulate rosmarinic acid accumulation in the cell cultures of *Satureja khuzistannica* [63] and anthocyanins in shoot cultures of *Prunus salicina* × *P. persica* [64], while SA stimulated the production of hypericin and pseudohypericin in the cell cultures of *Hypericum perforatum* [27].

In G. *lutescens* shoot cultures, all tested elicitors contributed almost equally to enhance accumulation of xanthones magniferin, DMB-8-*O*-glc, and bell-8-*O*-glc, while MeJA was superior for elicitation of xanthone aglycons DMB and bellidifolin (Figure 9B–F). The content of xanthone glucosides mangiferin, DMB-8-*O*-glc, and bell-8-*O*-glc at all applied concentration of JA was almost 2-fold higher than in control shoots. However, with the increase in MeJA concentration, production decreased to the level of non-elicited controls. When SA was applied for elicitation, increased concentration did not significantly affect the level of mangiferin, but reduced the content of DMB-8-*O*-glc and bell-8-*O*-glc.

In contrast to xanthone glucosides, the production of aglycones DMB and bellidifolin were not affected or only slightly enhanced upon treatment with lower concentrations of JA and SA. However, shoots elicited with the higher MeJA concentrations showed a significant increase in accumulation of both aglycones. The highest DMB and bellidifolin content (1.14 mg g^−1^ DW and 1.68 mg g^−1^ DW, respectively) was recorded at a concentration of 500 µM MeJA which was 7.4 and 7.6 times higher than in the control shoots, respectively.

A similar positive effect of elicitors on production of xanthone aglycones in hairy roots of *G. dinarica* was reported by Krstić-Milošević et al. [65], where application of biotic elicitors strongly increased production of aglycone norswertianin while simultaneously reducing the production of its glycoside norswertianin-1-*O*-primeveroside. These findings pointed out a selective accumulation of specific xanthone compounds upon treatment with elicitors. In *G. lutescens* shoots they suppressed glycosylation, the late step of the constitutive pathway, leading to increased accumulation of xanthone aglycones. High level of xanthone aglycones is important defensive mechanism for protection the plant cells from oxidative damage, as well as to impair pathogen growth.

Different observations have been recorded regarding the individual efficacy of the particular elicitor compounds. Elicitation effect of MeJA was higher in the production of gymnemic acid in cell suspension of *Gymnema sylvestre* [66], and iridoids and phenylethanoids in the hairy root culture of *Rehmannia glutinosa* [67]. MeJA has also been demonstrated to increase xanthone production in *H. perforatum* cell suspension cultures [25] and in combination with sucrose showed remarkable stimulating effects on hypericin and hyperforin production [68]. On the other hand, higher elicitation effect of SA compared to MeJA was reported for withanolides production from *Withania somnifera* [61]. SA was also more favorable for kinsenoside accumulation and had an equal effect on polysaccharide accumulation with MeJA during *A. roxburghii* rhizome culture [69].

However, many reports suggested that along with the choice of suitable elicitor compound its benefit for the production of metabolite of interest also depended on the concentration used and duration of elicitor treatment.

In *G. lutescens* shoots increased concentration of MeJA showed different effect on accumulation of xanthones. The highest MeJA dose (500 μM) led to the highest DMB and bellidifolin content, while on the other hand strongly reduced the production of xanthone glucosides. A similar effect of SA on accumulation of cardiac glycosides has been shown in shoot cultures of *Digitalis purpurea* [70]. Namely, addition of 50 μM SA promoted production of digitoxin, but increasing concentrations of SA drastically reduced its content. On the contrary, digoxin accumulation was increased in the shoots with an increase in SA concentration up to 200 μM. The flavonoid production of *H. perforatum* cell cultures was significantly promoted by 100 μM MeJA [26], while 4.46 μM MeJA increased the isoflavonoid content in *Pueraria mirifica* cell culture [71]. JA has also been reported as an effective elicitor of secondary metabolite production in many plant species. Thus, enhanced production of oleanolic acid was obtained in cell suspension cultures of *Calendula officinalis* by elicitation with 100 μM JA for 72 h [72], while supplementation of 50 μM JA on day 12 induced the highest anthraquinone content in cell suspension cultures of *Morinda elliptica* [73].

In summary, the findings of the current study suggest that the effectiveness of elicitation of secondary metabolites depends on several factors, including the type of elicitor, the elicitor concentration, the time of exposure to elicitor treatment, and the culture conditions [74].

## 3. Materials and Methods

### 3.1. Plant Material

*Gentianella lutescens* plants at fruitful stage were collected in September 2015 in their native habitat on the southern outskirts of the large mountain field Veliko Košlje at Povlen Mountain, in the locality of Razbojište (latitude 440 10′ 3.43″ north and longitudes 140 37′ 33.474″ east), the Republic of Serbia. A voucher specimen (Co. 6392113/04) was deposited in the Herbarium at the Natural History Museum, Belgrade.

### 3.2. In Vitro Seed Germination and Shoot Culture Initiation

Immature fruits harvested from collected plants were disinfected in 100 mL 20% (*v*/*v*) commercial bleach solution (4–5% NaOCl) with two drops of liquid detergent (Fairy^®^) for 25 min and then rinsed 3 times with sterile distilled water.

Immature seeds (1.5 mm) were isolated under stereomicroscope and germinated in Ø90 mm Petri dishes filled with 25 mL of germinating medium consisting of Murashige and Skoog’s (MS) mineral salts [75], LS vitamins [76], and 0.64% (*w*/*v*) agar (Institute of Virology, Vaccines and SeraTorlak, Belgrade, Serbia). For shoot culture initiation, the epicotyl explants were isolated from 7-day old seedlings from five seedling lines and placed onto shoot induction medium, which was basal medium (BM) consisting of MS mineral salts, LS vitamins, 100 mg L^−1^ myo-inositol (Sigma-Aldrich, Steinheim, Germany)), 58.4 mM sucrose with addition 0.2 mg L^−1^ BA. The medium was gelled with 0.64% (*w*/*v*) agar (Institute of Virology, Vaccines and SeraTorlak, Belgrade, Serbia). Individual epicotyl explant of each seedling line was cultured separately in 100-mL wide-neck Erlenmeyer flask with cotton-wool plugs. Obtained shoots were maintained on the same medium (BM + 0.2 mg L^−1^ BA) and subcultured every 5 weeks on fresh medium. For shoot multiplication, 8–10 shoots of the same line were cultured in Erlenmeyer flask containing basal medium with addition of BA at increasing concentrations 0, 0.05, 0.1, 0.2, 0.5, 1.0, and 2.0 mg L^−1^ for 5 weeks and the number of propagated shoots were recorded. Multiplication index was calculated as main shoot + new axillary shoots (length ≥ 2 mm).

For rooting, solid ½ MS medium supplemented with 2% sucrose and increasing (0.2, 0.5, 1.0, 2.0, and 5.0 mg L^−1^) IBA (indole-3-butyric acid, Sigma-Aldrich) was used. Different durations of root induction treatment were applied, lasting 1–6 days at 5 mg L^−1^ IBA to 50 days at 0.2–2.0 mg L^−1^ IBA (Table 3). For elongation of induced roots, the shoots were transferred into liquid PGR-free ½ MS medium in the test tubes 20 × 150 mm with metal holders containing ≈15 mL of liquid medium. Elongation treatments lasted from 52 to 66 days depending on the concentration of auxin applied (Table 3). Two replicates with 15–26 shoots per rooting treatment were performed and the percentage of rooted shoots and root features were estimated.

All media pH was adjusted to 5.8 prior to autoclaving for 25 min at 114 °C. Cultures were grown at 25 ± 2 °C in a controlled environment room illuminated with cool-white Phillips fluorescent lamps providing 35–45 μmol m^−2^ s^−1^ under a 16-h (long day) photoperiod.

### 3.3. Increase in Sucrose and Sorbitol Level

To examine the effect of enhanced osmotic pressure on the shoot growth and xanthone production, 400 mg of *G. lutescens* shoots were cultivated per Erlenmeyer flask on BM + 0.2 mg L^−1^ BA wherein sucrose or sorbitol at increasing concentrations 58.4, 116.8, 175.2, 233.6 mM were added. Shoot growth was measured after 5 weeks of culture in terms of fresh weight and dry weight of shoots per Erlenmeyer flask. Growth index [(final fresh weight − initial fresh weight)/initial fresh weight], % dry matter [(dry weight/final fresh weight) × 100], and accumulation of xanthones were determined for the harvested shoots.

### 3.4. Elicitor Preparation and Application

Jasmonic acid (JA, Duchefa), methyl jasmonate (MeJA, Duchefa) and salicylic acid (SA, Duchefa) 10 mM stock solutions were prepared in 50% (*v*/*v*) ethanol and then filter-sterilized using 0.22 μm filter. The shoots (400 mg weight) were isolated from 35–40-day-old cultures and grown in BM + 0.2 mg L^−1^ BA supplemented with elicitors at the following final concentrations: 100, 200, 300, and 500 μM. The treatment with elicitor lasted 7 days, and then the shoots were cultured on a control medium for another 7 days. Control shoots were cultured on BM + 0.2 mg L^−1^ BA without elicitors. At the end of the experiment growth index [(final fresh weight − initial fresh weight)/initial fresh weight] and accumulation of xanthones were determined for the harvested shoots.

### 3.5. Xanthone Extraction and Conditions for HPLC-DAD Analysis

Xanthone compounds extraction from dry powered plant material was performed as previously reported by Krstić-Milošević et al. [77]. Obtained extracts were filtered into volumetric flasks (10 mL) and adjusted to the volume with methanol. Prior to HPLC analysis, extracts were filtered through a nylon syringe filters (Captiva syringe filters, 0.45 mm, 13 mm, Agilent Technologies, Waldbronn, Germany). Chromatographic analysis was carried out on Agilent series 1100 HPLC instrument (Agilent Technologies, Waldbronn, Germany), with a diode array detector, on a reverse phase Zorbax SB-C18 (Agilent Technologies, Waldbronn, Germany) analytical column (150 mm × 4.6 mm i.d., 5 μm particle size) thermostated at 25 °C. The mobile phase consisted of two solvents: A (1% *v*/*v* solution of orthophosphoric acid in water) and B (acetonitrile) in the following gradient elution: 98–90% A 0–5 min, 90% A 5–13 min, 90–75% A 13–15 min, 75% A 15–18 min, 75–70% A 18–20 min, 70–40% A 20–24 min, 40–0% A 24–28 min. The injection volume was 5 μL. Detection wavelengths were set at 260 and 320 nm, and the flow rate was 1 mL min^−1^. The isolation, identification, and characterization of xanthones demethylbellidifolin-8-*O*-glucoside (DMB-8-*O*-glc), bellidifolin-8-*O*-glucoside (bell-8-*O*-glc), demethylbellidifolin (DMB) and bellidifolin were reported in previous study [40]. Mangiferin was purchased from Sigma-Aldrich (Steinheim, Germany). Quantification was performed using standardized calibration curves of xanthones. The content of xanthones in the samples was determined by calculation of peak area and expressed as milligrams per gram of dry weight.

### 3.6. Statistical Analysis

All in vitro culture experiments were repeated at least 2–4 times with 15–40 shoots/explants per treatment. Biochemical analyses were repeated 4–11 times. The data were subjected to standard one-way analysis of variance (ANOVA) except the data related to the effects of elicitors where two-way ANOVA was applied. Percentage data were subjected to angular transformation before statistical analysis, followed by inverse transformation for presentation. Means and standard errors were calculated for numerical parameters and their differences was analyzed by *t*-test or Fisher’s multiple range LSD test at *p* ≤ 0.05 using the StatGraphics Plus software package for Windows 2.1 (Statistical Graphics Corp., Rockville, MD, USA).

## 4. Conclusions

The present study represents the first achievement in establishing an in vitro culture of *G. lutescens* that can be used for mass shoot propagation, significantly contributing to the preservation of this valuable medicinal plant. This is also the first report on the composition of pharmacologically active xanthones in wild and tissue cultured shoots of *G. lutescens.* The promising in vitro protocol included sterile germination of immature seeds on PGR-free MS medium, establishment of shoot culture from epicotyl explants isolated from seedlings on MS + 0.2 mg L^−1^ BA, as well as shoot multiplication and elongation on the same substrate. Satisfactory rooting included two phases, root induction on ½ MS medium with 5.0 mg L^−1^ IBA for 2 days, followed by root elongation in liquid PGR-free ½ MS medium for 60 days. The results indicated that the use of increased sucrose, as well as abiotic elicitors JA, MeJA and SA stimulated the accumulation of bioactive xanthones in tissue-cultured shoots. Multiple increases of the amount of aglycones BMD and bellidifolin was achieved by applying MeJa at the highest concentration. Since in vitro shoot cultures of *G. lutescens* were proved as a prospective modality for the accumulation of xanthones of significant interest, advanced conditions will be applied in further studies for the production of metabolites in large-scale in bioreactors. These achievements increase the commercial prospects of *G. lutescens* in the medicinal plant industry.

## Figures and Tables

**Figure 1 plants-10-01651-f001:**
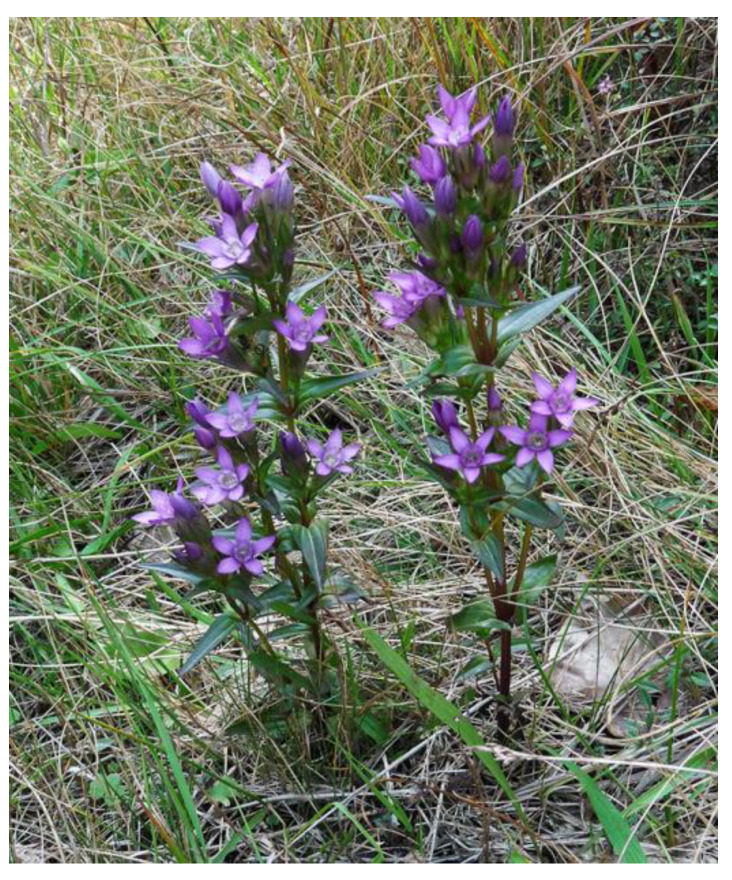
*Gentianella lutescens* subsp. *carpatica* in a natural habitat on Povlen mountain (locality Razbojište), Serbia.

**Figure 2 plants-10-01651-f002:**
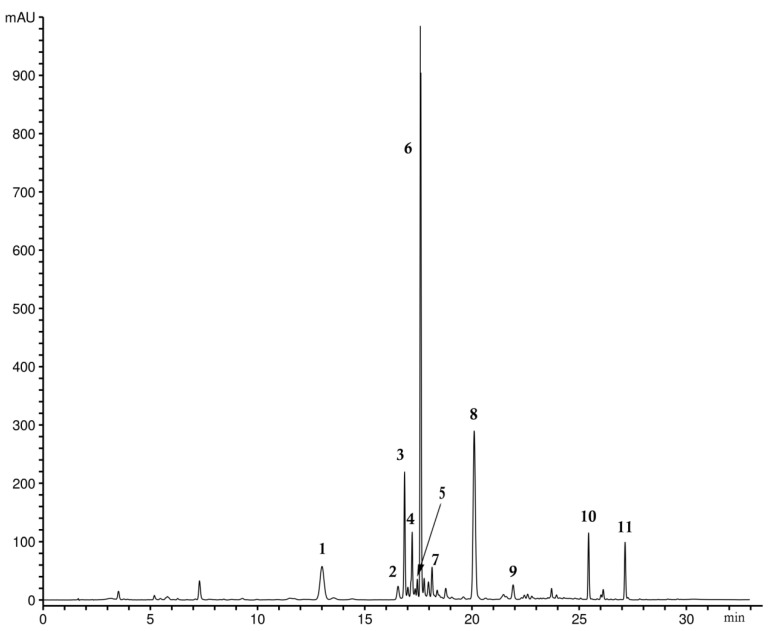
HPLC profile (λ = 260 nm) of methanol extract of *G. lutescens*. Peaks: 1—swertiamarin, 2—gentiopicrin, 3—mangiferin, 4—campestroside, 5—isoorientin, 6—demethylbellidifolin-8-*O*-glucoside, 7—swertisin, 8—bellidifolin-8-*O*-glucoside, 9—veratriloside, 10—demethylbellidifolin, 11—bellidifolin.

**Figure 3 plants-10-01651-f003:**
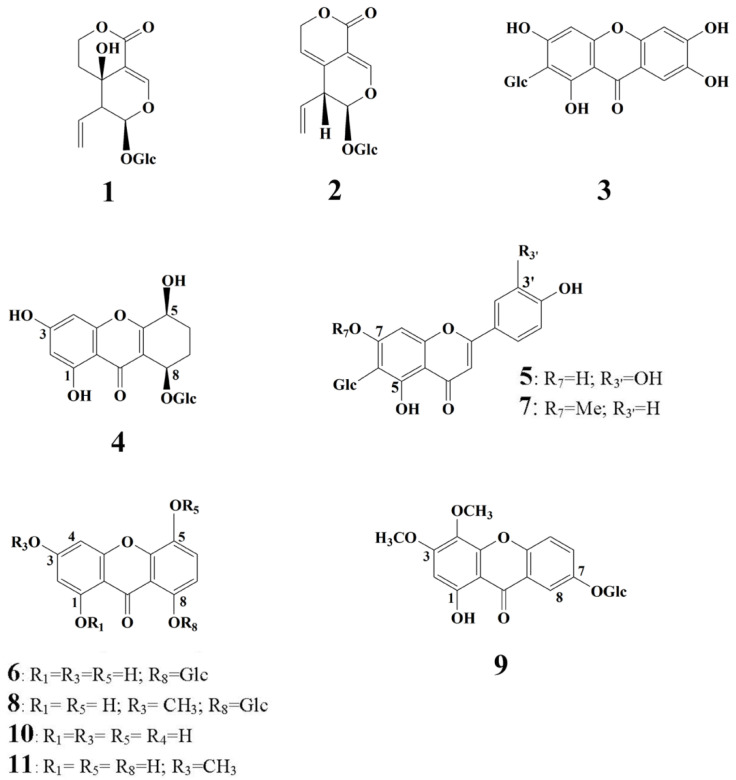
Chemical structures of secoiridoid and xanthone compounds identified in *G. lutescens*. **1**—swertiamarin, **2**—gentiopicrin, **3**—mangiferin, **4**—campestroside, **5**—isoorientin, **6**—demethylbellidifolin-8-*O*-glucoside, **7**—swertisin, **8**—bellidifolin-8-*O*-glucoside, **9**—veratriloside, **10**—demethylbellidifolin, **11**—bellidifolin.

**Figure 4 plants-10-01651-f004:**
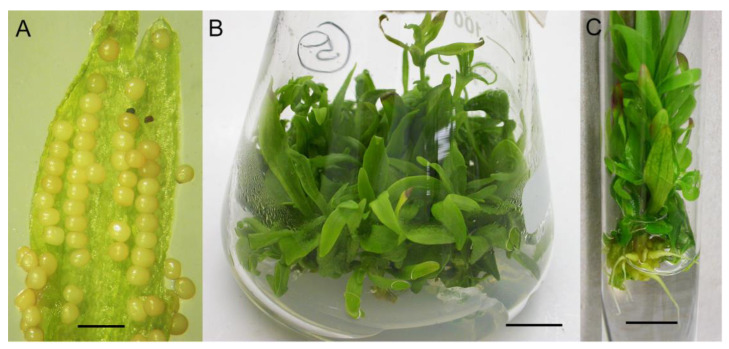
In vitro propagation of *G. lutescens*. (**A**) Open pod with immature seeds (*bar* = 5 mm), (**B**) shoot multiplication on BM + 0.2 mg L^−1^ BA (*bar* = 10 mm), (**C**) root elongation in PGR-free ½ MS liquid medium after treatment of shoots with 5.0 mg L^−1^ IBA for 2 days (*bar* = 10 mm).

**Figure 5 plants-10-01651-f005:**
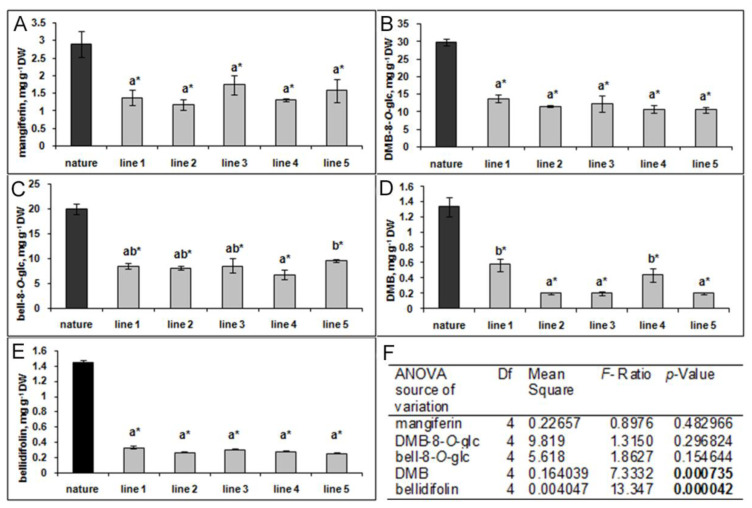
The content of xanthones mangiferin (**A**), demethylbellidifolin-8-*O*-glucoside (DMB-8-*O*-glc) (**B**), bellidifolin-8-*O*-glucoside (bell-8-*O*-glc) (**C**), demethylbellidifolin (DMB) (**D**) bellidifolin (**E**) in five shoot lines of *G. lutescens* cultured on BM + 0.2 BA mg L^−1^ for 35 days. Wild *G. lutescens* herb grown in nature was used as a control. Values are the means ± SE of four to six biological replicates *(n* = 4–6). Data of xanthones content in shoot lines were analyzed by one-way ANOVA (**F**). The values followed by different letters were significantly different according to Fisher’s LSD test at *p* ≤ 0.05; asterisk (*) indicate a significant difference of values between in vitro shoot lines and wild plants samples according to Student’s *t*-test at *p* ≤ 0.05.

**Figure 6 plants-10-01651-f006:**
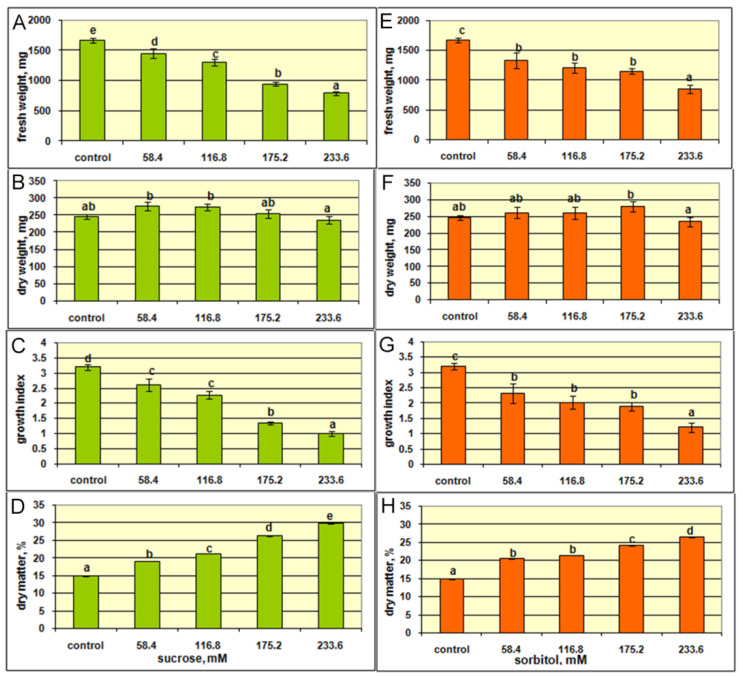
The effect of increasing concentrations (58.4, 161.8, 175.2, and 233.6 mM) of sucrose (**A**–**D**) and sorbitol (**E**–**H**) on the growth of shoot cultures of *G. lutescens* (line 5) after 35 days of culture. The increasing concentrations of sucrose or sorbitol were added into control medium (BM + 0.2 mg L^−1^ BA containing 58.4 mM sucrose). Values are the means ± SE of eight to ten biological replicates (*n* = 8–10). Data were analyzed by one-way ANOVA. Values followed by different letters are significantly different according to Fisher’s LSD test at *p* ≤ 0.05.

**Figure 7 plants-10-01651-f007:**
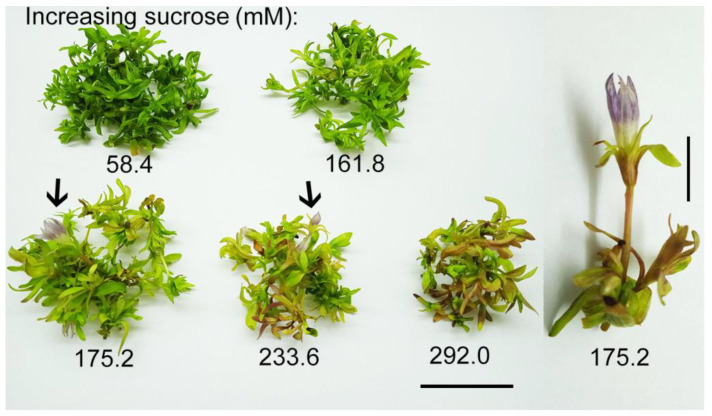
The effect of sucrose at increasing concentrations (58.4, 161.8, 175.2, and 233.6 mM) on the growth and flowering of shoot cultures of *G. lutescens* (line 5) after 35 days of culture (*bar* = 20 mm). The increasing concentrations of sucrose were added into control medium (BM + 0.2 mg L^−1^ BA containing 58.4 mM sucrose). Arrows indicate floral bud development. Right—shoot with normally developed flower regenerated on the medium with 175.2 mM sucrose (*bar* = 5 mm).

**Figure 8 plants-10-01651-f008:**
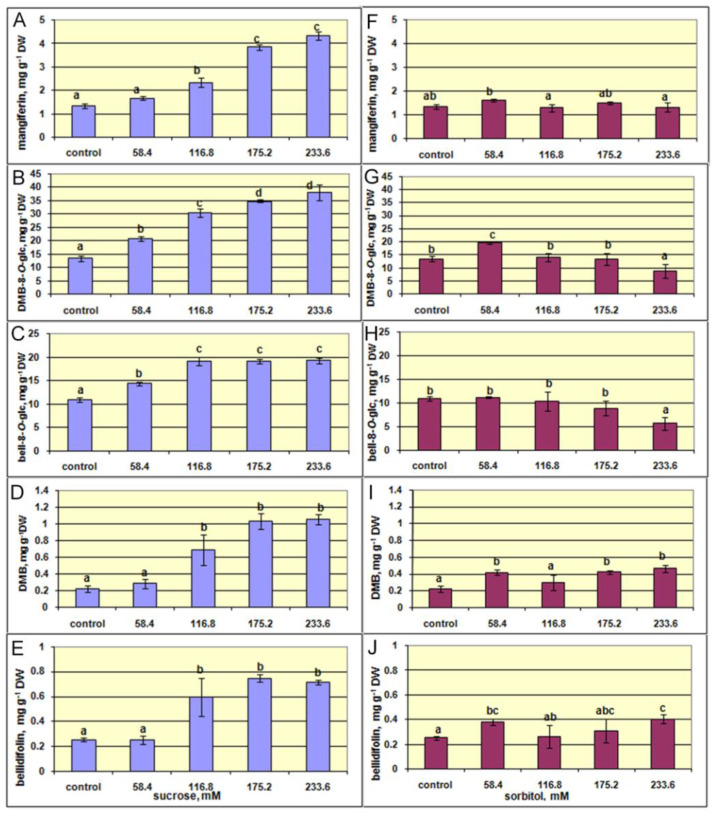
The effect of increasing concentrations (58.4, 161.8, 175.2, and 233.6 mM) of sucrose (**A**–**E**) and sorbitol (**F**–**J**) on the content of xanthones mangiferin, demethylbellidifolin-8-*O*-glucoside (DMB-8-*O*-glc), bellidifolin-8-*O*-glucoside (bell-8-*O*-glc), demethylbellidifolin (DMB) and bellidifolin in shoots cultures of *G. lutescens* (line 5) after 35 days of culture. The increasing concentrations of sucrose or sorbitol were added into control medium (BM + 0.2 mg L^−1^ BA containing 58.4 mM sucrose). Values are the means ± SE of four to eleven biological replicates (*n* = 4–11). Values denoted by the same letter are not significantly different according to the Fisher’s LSD test at *p* ≤ 0.05 following one-way ANOVA.

**Figure 9 plants-10-01651-f009:**
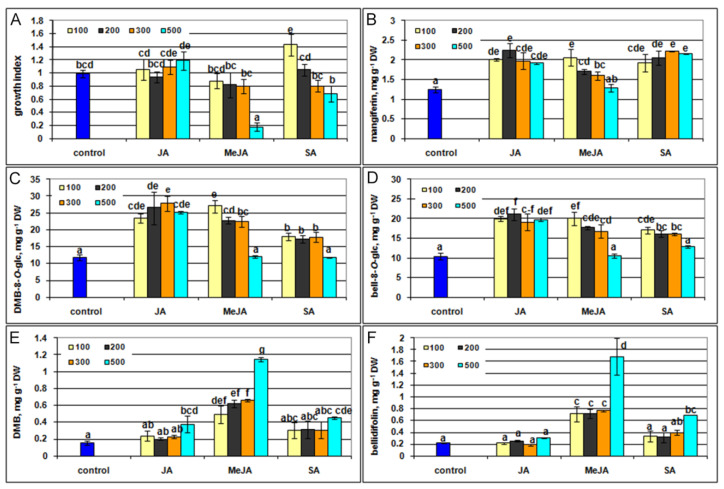
Effect of elicitors jasmonic acid (JA), methyl jasmonat (MeJA) and salicilic acid (SA) at increasing concentrations (100, 200, 300, and 500 µM) on the growth index and xanthone production of shoot cultures of *G. lutescens* (line 5). Growth index (**A**), mangiferin (**B**), DMB-8-*O*-glc (**C**), bell-8-*O*-glc (**D**), DMB (**E**), and bellidifolin (**F**) content. Data represent means ± SE of four to seven biological replicates (*n* = 4–7). Values denoted by the same letter are not significantly different according to the Fisher’s LSD test at *p* ≤ 0.05 following ANOVA multifactorial analysis.

**Table 1 plants-10-01651-t001:** Shoot induction and multiplication and the length of the main shoot in *G. lutescens*. Epicotyl explants of five seedling lines were cultivated on BM + 0.2 mg L^−1^ BA for 35 days. Values represent the means ± SE from two experiments with 35–40 samples per line (*n* = 75–80). Data were analyzed by one-way ANOVA analysis. Within each column means followed by different letters are significantly different according to Fisher’s LSD test at *p* ≤ 0.05. Multiplication index: main shoot + axillary buds.

**Line** **(BA 0.2 mg L^−1^)**	**No. of Explants**	**Multiplication Index ± SE**	**Length of Main Shoot (mm) ± SE**	
line 1	80	3.03 ± 0.23 ab	14.49 ± 0.65 a	
line 2	77	3.31 ± 0.19 ab	17.40 ± 0.92 ab	
line 3	80	3.53 ± 0.23 b	23.05 ± 1.67 c	
line 4	75	2.76 ± 0.25 a	17.19 ± 1.22 ab	
line 5	80	3.41 ± 0.26 ab	19.39 ± 1.07 b	
**ANOVA** **Source of Variation**	**Df**	**Mean Square**	***F*-Ratio**	***p*-Value**
Multiplication index	4	7.48476	1.73	0.1428
Length of main shoot	4	804.293	7.67	0.0000

**Table 2 plants-10-01651-t002:** Effect of increasing concentrations (0–2.0 mg L^−1^) of BA on shoot multiplication, the length of the main shoot and vitrification incidence of *G. lutescens* line 5. Values represent the means ± SE from 3–4 experiments with 25–40 samples per treatment (*n* = 80–129). Data were analyzed by one-way ANOVA analysis. Within each column means followed by different letters are significantly different according to Fisher’s LSD test at *p* ≤ 0.05. Results were scored after 35 days. Multiplication index—main shoot + axillary buds.

**BA** **(mg L^−1^)**	**No. of Expl.**	**Multiplication Index ± SE**	**Length of Main** **Shoot (mm) ± SE**	**Vitrification**	
				**No.**	**%**
0	85	2.89 ± 0.16 a	15.02 ± 0.66	1	0.8
0.05	100	3.16 ± 0.17 a	15.85 ± 0.50 e	1	1.0
0.1	128	3.20 ± 0.14 a	13.66 ± 0.46 cd	7	5.47
0.2	90	3.92 ± 0.25 b	15.48 ± 0.64 cd	10	7.75
0.5	129	3.22 ± 0.17 a	13.09 ± 0.48 bc	24	18.6
1.0	128	2.88 ± 0.15 a	12.09 ± 0.38 b	20	15.6
2.0	80	2.94 ± 0.20 a	10.46 ± 0.47 a	14	17.5
**ANOVA** **Source of Variation**	**Df**	**Mean Square**	***F*-Ratio**	***p*-Value**	
Multiplication index	6	12.2028	3.72	0.0012	
Length of main shoot	6	351.866	12.96	0.0000	

**Table 3 plants-10-01651-t003:** Effect of pretreatments with IBA on the rooting of shoots of *G. lutescens* line 5. After IBA treatment the shoots were cultivated in liquid PGR-free ½ MS medium with 2% sucrose and fresh medium was added every 7 days. Experiments were repeated 2 times with 15–26 shoots per treatment (*n* = 30–52). Within each group of experiments (distinct color) means followed by different letters in the column are significantly different according to Fisher’s LSD test at *p* ≤ 0.05. SE—standard error.

IBA (mg L^−1^)	Auxin Treatment + PGR-Free Liquid Medium = Duration of Experiment (Day)	Explants No.	Shoot with Root Primordia (%)	Rooting (%)	Roots Per Rooted Explant ± SE	Length of the Longest Root (mm) ± SE
0.2	50 + 55 = 105	40	12.5	17.5	2.57 ± 0.84 a	7.29 ± 1.49 a
0.5	50 + 55 = 105	52	13.5	7.7	2.25 ± 0.48 a	8.75 ± 2.56 a
1.0	50 + 55 = 105	50	16.0	14.0	3.43 ± 0.95 a	5.86 ± 2.41 a
2.0	50 + 55 = 105	50	6.0	6.0	2.33 ± 0.33 a	10.51 ± 7.51 a
1.0	14 + 66 = 80	34	8.8	0	-	-
1.0	21 + 59 = 80	37	8.1	3.3	2.0 ± 0	2.0 ± 0
1.0	28 + 52 = 80	45	13.3	0	-	-
2.0	14 + 66 = 80	30	16.7	6.7	4.5 ± 0.5 a	14.0 ± 2.0 a
2.0	21 + 59 = 80	30	16.7	13.3	5.25 ± 1.1 a	13.75 ± 4.27 a
2.0	28 + 52 = 80	33	6.1	0	-	-
5.0	1 + 61 = 62	30	13.3	30.0	2.89 ± 0.5 a	12.4 ± 2.02 a
5.0	2 + 61 = 63	30	23.3	33.3	5.4 ± 1.1 b	11.2 ± 2.6 a
5.0	4 + 61 = 65	30	16.7	33.3	3.5 ± 0.6 ab	7.7 ± 1.23 a
5.0	6 + 59 = 65	30	10.0	16.7	2.4 ± 1.2 a	7.6 ± 1.21 a

## Data Availability

The data presented in this study are available from the authors.

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
