# Peer review of "Gentianella lutescens subsp. carpatica J. Holub.: Shoot Propagation In Vitro and Effect of Sucrose and Elicitors on Xanthones Production"

_plants, 2021, doi:10.3390/plants10081651_

Round 1
Reviewer 1 Report
The manuscript entitled “Gentianella lutescens subsp. carpatica J. Holub.: Shoot Propagation in vitro and Effect of Sucrose and Elicitors on Xanthones Production” provides an exhaustive study about the establishment and manipulation of G. lutescens in vitro cultures. The authors studied the effects of different in vitro conditions and their effects on shooting, rooting, and secondary metabolite production. I consider that the work will provide valuable information that can serve as the basis for in vitro manipulation of G. lutescens. Nevertheless, some points need to be addressed to improve the manuscript to accomplish some of the scientific standards, ensuring the reproducibility of the results. In the next part of this document, I will specify the different comments that need to be addressed.
- In the introduction, some extra information about the biosynthetic pathway of the compounds studied needs to be added, this will help to the interpretation of the results.
- The HPLC-DAD analysis (Line 132) is not explained in materials and methods.
- The quality of figure 3 needs to be improved.
- The reference of all the mediums used needs to add.
- In line 197, why line 5 was selected? The readers need to know which reason motivates all the steps of the manuscript.
- Lines 206-209: The authors talk about the results of maintaining the cultures for more than 1 year. These results need to be discussed in more detail, or with more data that can support the conclusions provided by the authors.
- In table 3, the different days corresponding to auxin treatment are specified. Why with 5 mg/L of IBA the authors use only 1 to 6 days? This change needs to be discussed since this experimental design differs from the main assay (14-21-28 days in IBA).
- In Line 435, why 7 days + 7 days was chosen? Add references.
- As a general comment, when the authors talk about the results in other Gentianella sp., they must specify the protocols that the original work was used in terms of treatments, time, concentrations, etc. (Line 308-309: which concentrations? etc).
- In Figure 9, the colors of the bars are different between panels.
- There are several typos during the manuscript, check carefully (in vitro without italics, supplementary (Line 325), Naimely (Line 309), etc).
- A final table/summary of the selected conditions for the authors like optimal for in vitro culture/rooting/production needs to be added as the conclusion.
- In general, English needs to be corrected, since there are some mistakes. For example, “Thus” is used incorrectly in several parts of the manuscript.
Author Response
Dear Reviewer,
Thank you for your kind letter of decision on our manuscript “Gentianella lutescens subsp. carpatica J. Holub.: Shoot Propagation in vitro and Effect of Sucrose and Elicitors on Xanthones Production ”. We appreciate your time and efforts in reviewing this manuscript and give our thanks for your comments and suggestions.
Your comments have motivated us to explain briefly our study. We have applied corrections in the content of this manuscript according to your suggestions. We believe that these changes made certain improvement of this paper and contribute to better understanding of the particular responses to the comments. In revision we have taken all of the reviewers’ comments into account. Here are our answers to specific points:
The manuscript entitled “Gentianella lutescens subsp. carpatica J. Holub.: Shoot Propagation in vitro and Effect of Sucrose and Elicitors on Xanthones Production” provides an exhaustive study about the establishment and manipulation of G. lutescens in vitro cultures. The authors studied the effects of different in vitro conditions and their effects on shooting, rooting, and secondary metabolite production. I consider that the work will provide valuable information that can serve as the basis for in vitro manipulation of G. lutescens. Nevertheless, some points need to be addressed to improve the manuscript to accomplish some of the scientific standards, ensuring the reproducibility of the results. In the next part of this document, I will specify the different comments that need to be addressed.
Response: Thank you very much for your kind words about our paper. We are grateful for the energy and the time you expended on our behalf.
- In the introduction, some extra information about the biosynthetic pathway of the compounds studied needs to be added, this will help to the interpretation of the results.
Response: Thank you for this suggestion. We have made the requested changes in the Introduction section.
- The HPLC-DAD analysis (Line 132) is not explained in materials and methods.
Response: We are sorry that we did not use the uniform term “HPLC-DAD” for chromatographic analysis throughout the manuscript. That has been corrected and HPLC-DAD analysis is explained in M&M section.
- The quality of figure 3 needs to be improved.
Response: Requested correction has been made.
- The reference of all the mediums used needs to add.
Response: The reference for all medium used has been inserted in the text.
- In line 197, why line 5 was selected? The readers need to know which reason motivates all the steps of the manuscript.
Response: The reason is explained in the text.
Since there were no significant differences in the content of secondary metabolites and the growth index between G. lutescens hairy root lines, the Line 5 was chosen for the shoot proliferation experiment because it showed growth and multiplication stability over time and produced a large amount of healthy-looking shoots needed to start the experiment.
- Lines 206-209: The authors talk about the results of maintaining the cultures for more than 1 year. These results need to be discussed in more detail, or with more data that can support the conclusions provided by the authors.
Response: These results have been discussed in the text.
- In table 3, the different days corresponding to auxin treatment are specified. Why with 5 mg/L of IBA the authors use only 1 to 6 days? This change needs to be discussed since this experimental design differs from the main assay (14-21-28 days in IBA).
Response: IBA treatment duration has been disused in the text.
Decades of experience in working with gentians and other plant species, as well as the literature data, have indicated that with increasing auxin concentration in the growth medium the length of induction treatment should be shortened, otherwise callus would be formed instead of roots. Thus, with 50 mg/L IBA treatment lasts a maximum 24 hours, when 5 and 10 mg / L IBA was used treatment lasts maximum 6 days, with concentrations up to 2 mg / L IBA treatment lasts the entire passage. The results of rooting of G. lutescens shoots confirmed previously statement that prolonged exposure to higher auxin concentrations mainly leads to plantiful callus development. Experimental design and the main assay have been aligned in the text.
According to suggestion of the Reviewer # 2 a large part of the Result section related to rooting has been shortened and the repetitions have been avoided.
- In Line 435, why 7 days + 7 days was chosen? Add references.
Response: Plant tissue reacts relatively quickly to elicitation depending on the type of explant and the type of substrate (liquid or solid), to induce the synthesis of secondary metabolites for several hours to several days (Wungsintaweekul J, et al. 2012. Ziaratnia S et al 2009). Prolonged exposure to an elicitor even reduces the accumulation of the secondary metabolite over time. Thus, in 2013, Sharma et al obtained the highest production of bacoside in shoots of Bacopa monnieri in the first 7 days of elicitation with MeJA, and, Munish et al. 2015. with the same type by elicitation with JA.
The return of shoots to the substrate without elicitor an additional 7 days before material collection was also used by Sivanandhan, G. et al 2013, after elicitation of Withania somnifera shoots with SA and MeJA.
The experienced references were added.
- Sivanandhan, G.; Rajesh, M.;Arun, M.; Jeyaraj, M;Kapil Dev, G.; Arjunan, A.; Manickavasagam, M.;Muthuselvam, M.; Selvaraj, N.; Ganapathi, A. Effect of culture conditions, cytokinins, methyl jasmonate and salicylic acid on the biomass accumulation and production of withanolides in multiple shoot culture of Withania somnifera (L.) Dunal using liquid culture. Acta Physiol. Plant. 2013, 35, 715-728.
- Sharma, P.; Yadav, S.; Srivastava, A.; Shrivastava, N. Methyl jasmonate mediates upregulation of bacoside A production in shoot cultures of Bacopa monnieri.
Lett. 2013, 35,1121-1125. - Munish S, Ashok A, Rajinder G, Sharada M. Enhanced bacoside production in shoot
cultures of Bacopa monnieri under the influence of abiotic elicitors. Natural Product
2015;29(8):745–749. - Wungsintaweekul J, Choo-malee J, Charoonratana T, Niwat N (2012) Methyl jasmonate and yeast extract stimulate mitragynine production in Mitragyna speciosa (Roxb.) Korth shoot culture. Biotechnol Lett 34:1945–1950
- Ziaratnia S, Kunert K, Lall N. 2009. Elicitation of 7-methyljuglone in Drosera capensis Afr J Bot. 75:97–103.
- As a general comment, when the authors talk about the results in other Gentianella sp., they must specify the protocols that the original work was used in terms of treatments, time, concentrations, etc. (Line 308-309: which concentrations? etc).
Response: Thank you for this observation. It is specified as you suggested.
- In Figure 9, the colors of the bars are different between panels.
Response: In the revised figure 9 the colours of the bars are uniform.
- There are several typos during the manuscript, check carefully (in vitro without italics, supplementary (Line 325), Naimely (Line 309), etc).
Response: The typos are corrected throughout the manuscript. We are sorry that we made such mistakes. Thank you.
- A final table/summary of the selected conditions for the authors like optimal for in vitro culture/rooting/production needs to be added as the conclusion.
Response: A final summary related to selected in vitro protocol is added as the conclusion.
- In general, English needs to be corrected, since there are some mistakes. For example, “Thus” is used incorrectly in several parts of the manuscript.
Response: English is corrected as well as the mistakes throughout the manuscript.
With my best regards,
On behalf of all co-authors,
Sincerely
Dijana Krstic- Milosevic
Reviewer 2 Report
It seems that the authors have a good competence with Gentianella sp. as evidenced by the self citation (8 paper for Krstić-Miloševic and 6 for Vinterhalter, B) . In fact the authirs have already used micropropagation and in vitro culture for the production of secondary metabolites of various species of Gentianella with the aim to produce the xanthone, a useful compound contrasting various beneficial pharmacological activities and also used as insecticide (please add reference for this use).
So other papers are already published with similar goals and methodology. Therefore the manuscript sounds to lack of novelty, although a conspicuous number of experiments have been conducted to highlight the aims of the papers:
“evaluate in vitro grown shoot culture of G. lutescens as an alternative, sustainable and stable source of xanthones. In order to increase biomass and xanthones content in G. lutescens shoots, the effect of sucrose, sorbitol and elicitors JA, MeJA, and SA, on shoot growth and xanthones production was investigated.
I invite the authors to underline the relevant novelty of some results, and reduce the part of micropropation technique.
INTRODUCTION
This part must be reduced, expecially
-the lines 84-93, well established the importance of biotechnology for the domestication of medicinal plants
- lines 98-104 : general use of in vitro culture
- lines 108-119 . reduce the part of the use of elicitors
RESULTS AND DISCUSSION
Table 1: it is not clear the number of samples replication, experiments. Please specify better.
Are 2 or or 3 experiments? the number of samples are 25 or 40 ( which experiment 25 and which 40) and the replications were 2 or 3 each sample?
Line 175-180 are repetition of methodology: delete
Line258-290 : reduce redundant and repetitive
Line 307 have and not has
Line 309 namely , not naimely
Lines354-373 es: comment of figure 6 which is the important difference in the addition of sucrose instead of sorbitol. no different patterns are showed
Figure 9: specify in the legend that each elicitor have been used with different concentration (100-500 μM)
line308 absent not apsent
give more detailed information and commentson the different elicitor results
CONCLUSION
lines 588- 604 are summary of results, not general conclusion of the work with perspective for future research
MATERIALS AND METHODS
which period (month)of year were plants harvested?
Author Response
Dear Reviewer,
Thank you for your kind letter of decision on our manuscript “Gentianella lutescens subsp. carpatica J. Holub.: Shoot Propagation in vitro and Effect of Sucrose and Elicitors on Xanthones Production ”. We appreciate your time and efforts in reviewing this manuscript and give our thanks for your comments and suggestions.
Your comments have motivated us to explain briefly our study. We have applied corrections in the content of this manuscript according to your suggestions. We believe that these changes made certain improvement of this paper and contribute to better understanding of the particular responses to the comments. In revision we have taken all of the reviewers’ comments into account. Here are our answers to specific points:
It seems that the authors have a good competence with Gentianella sp. as evidenced by the self citation (8 paper for Krstić-Miloševic and 6 for Vinterhalter, B) . In fact the authirs have already used micropropagation and in vitro culture for the production of secondary metabolites of various species of Gentianella with the aim to produce the xanthone, a useful compound contrasting various beneficial pharmacological activities and also used as insecticide (please add reference for this use).
- So other papers are already published with similar goals and methodology. Therefore the manuscript sounds to lack of novelty, although a conspicuous number of experiments have been conducted to highlight the aims of the papers:
- “evaluate in vitro grown shoot culture of lutescensas an alternative, sustainable and stable source of xanthones. In order to increase biomass and xanthones content in G. lutescens shoots, the effect of sucrose, sorbitol and elicitors JA, MeJA, and SA, on shoot growth and xanthones production was investigated.
- I invite the authors to underline the relevant novelty of some results, and reduce the part of micropropation technique.
Response: We appreciate this observation.
As suggested, the novelty of results is pointed up and the part of micropropagation is reduced.
INTRODUCTION
This part must be reduced, expecially
- -the lines 84-93, well established the importance of biotechnology for the domestication of medicinal plants
- - lines 98-104 : general use of in vitro culture
- - lines 108-119 . reduce the part of the use of elicitors
Response: All designated parts were reduced according to reviewer’s suggestions.
RESULTS AND DISCUSSION
- Table 1: it is not clear the number of samples replication, experiments. Please specify better.
- Are 2 or or 3 experiments? the number of samples are 25 or 40 ( which experiment 25 and which 40) and the replications were 2 or 3 each sample?
Response: The mistakes are corrected properly, Thank you! Two experiments were performed with 35-40 samples per line (n=75-80).
- Line 175-180 are repetition of methodology: delete
Response: Requested correction has been made in the text.
- Line258-290 : reduce redundant and repetitive
Response: A large part of the Result section has been shortened in the text, and repetitions have been avoided.
- Line 307 have and not has
Response: Thank you for pointing out this mistake. It is corrected in the text.
- Line 309 namely , not naimely
Response: Thank you for pointing out this mistake. It is corrected in the text.
- Lines354-373: comment of figure 6 which is the important difference in the addition of sucrose instead of sorbitol. no different patterns are showed
Response: If a two-factorial ANOVA was performed, it indicated that the type of carbohydrate added did not affect the growth of the shoots, but significantly affected in vitro flowering and dry matter content in the shoots. However, the concentrations of carbohydrates significantly affected all parameters at the same trend (Supplementary material, Table S2).
Supplementary material, Table S2. Two-way ANOVA of the influence of carbohydrate type (sorbitol or sucrose) and their concentrations on the in vitro growth and flowering of G. lutescens shoots.
Source |
Sum of Squares |
df |
Mean Square |
F-Ratio |
p-Value |
Fresh weight, mg |
|
|
|
|
|
(A) carbohydrate type |
3074 |
1 |
3074 |
0.102 |
0.750739 |
(B) concentration |
7471486 |
4 |
1867872 |
61.730 |
0.000000 |
A x B |
359189 |
4 |
89797 |
2.968 |
0.024209 |
Error |
2481220 |
82 |
30259 |
|
|
Dry weight,mg |
|
|
|
|
|
(A) carbohydrate type |
31 |
1 |
31 |
0.023 |
0.878964 |
(B) concentration |
15695 |
4 |
3924 |
2.994 |
0.023273 |
A x B |
5327 |
4 |
1332 |
1.016 |
0.403927 |
Error |
|
|
|
|
|
Growth index |
|
|
|
|
|
(A) carbohydrate type |
0.1364 |
1 |
0.1364 |
0.687 |
0.409736 |
(B) concentration |
49.3714 |
4 |
12.3429 |
62.145 |
0.000000 |
A x B |
1.7203 |
4 |
0.4301 |
2.165 |
0.080115 |
Error |
16.2864 |
82 |
0.1986 |
|
|
Dry matter, % |
|
|
|
|
|
(A) carbohydrate type |
16.19 |
1 |
16.19 |
7.10 |
0.009299 |
(B) concentration |
1998.38 |
4 |
499.60 |
219.03 |
0.000000 |
A x B |
64.24 |
4 |
16.06 |
7.04 |
0.000063 |
Error |
187.04 |
82 |
2.28 |
|
|
Flowering |
|
|
|
|
|
(A) carbohydrate type |
17.60000 |
1 |
17.60000 |
35.37255 |
0.000000 |
(B) concentration |
7.74417 |
4 |
1.93604 |
3.89107 |
0.006065 |
A x B |
7.74417 |
4 |
1.93604 |
3.89107 |
0.006065 |
Error |
|
|
|
|
|
- Figure 9: specify in the legend that each elicitor have been used with different concentration (100-500 μM)
Response: It is specified in the legend of Fig 9, as suggested .
- line308 absent not apsent
Response: Thank you for pointing out this mistake. It is corrected in the text.
- give more detailed information and comments on the different elicitor results
Response: More detailed information and comments related to different elicitors used are provided in the text. Thank you for this suggestion.
- CONCLUSION
lines 588- 604 are summary of results, not general conclusion of the work with perspective for future research
Response: Conclusion is corrected and rewritten according to suggestions of the reviewers.
- MATERIALS AND METHODS
which period (month)of year were plants harvested?
Response: The harvesting month is indicated in M&M
With my best regards,
On behalf of all co-authors,
Sincerely
Dijana Krstic- Milosevic
Round 2
Reviewer 1 Report
I consider that the authors did an extensive and deep improvement of the manuscript following the comments of the previous revision process, so the manuscript is now, from my point of view, high quality and suitable to be published in the present form.
Author Response
Dear Reviewer,
Thank you very much for your kind words about our paper.
Best regards,
D. Krstic-Milosevic
Reviewer 2 Report
From the editorial version (Pdf format) of the manuscript it looks like the authors have improved a former manuscript. However, in the editorial version some mistakes are found , that must be deleted. check the English language, because some sentences are difficult to read.
CONCLUSION: in part is a repetition. of the results (lines753- 773) . the conclusion must be rewritten without numbers, only emphasizing the results and the future perspective.
Author Response
Dear Reviewer,
We have applied some corrections in the manuscript. An English proofreading was done. However, according to the request of the Reviewer 1, we have added conditions for optimal in vitro culture/rooting/production of sec. metabolites in the conclusion. We have now made some changes to the conclusion according to your suggestions. We hope that these changes will be acceptable.
Best regards,
Dijana Krstic-Milosevic